# Radiomics Prediction of Muscle Invasion in Bladder Cancer Using Semi-Automatic Lesion Segmentation of MRI Compared with Manual Segmentation

**DOI:** 10.3390/bioengineering10121355

**Published:** 2023-11-25

**Authors:** Yaojiang Ye, Zixin Luo, Zhengxuan Qiu, Kangyang Cao, Bingsheng Huang, Lei Deng, Weijing Zhang, Guoqing Liu, Yujian Zou, Jian Zhang, Jianpeng Li

**Affiliations:** 1Department of Radiology, The Tenth Affiliated Hospital of Southern Medical University (Dongguan People’s Hospital), Dongguan 523059, China; yeyj0769@163.com (Y.Y.); mia_lei31@163.com (L.D.); Zouyujian@sohu.com (Y.Z.); 2Medical AI Lab, School of Biomedical Engineering, Shenzhen University Medical School, Shenzhen University, Shenzhen 518060, China; 2100241004@email.szu.edu.cn (Z.L.); qiuzx2022@mail.sustech.edu.cn (Z.Q.); 13266082905@163.com (K.C.); huangb@szu.edu.cn (B.H.); 3Imaging Department, Sun Yat-Sen University Cancer Center, State Key Laboratory of Oncology in South China, Collaborative Innovation Center for Cancer Medicine, Guangzhou 510060, China; zhangwj@sysucc.org.cn; 4College of Physics and Optoelectronic Engineering, Shenzhen University, Shenzhen 518060, China; liugq@szu.edu.cn; 5Shenzhen University Medical School, Shenzhen University, Shenzhen 518060, China; 6Shenzhen-Hong Kong Institute of Brain Science-Shenzhen Fundamental Research Institutions, Shenzhen 518060, China

**Keywords:** magnetic resonance imaging, urinary bladder neoplasms, radiomics, muscles, semi-automatic segmentation

## Abstract

Conventional radiomics analysis requires the manual segmentation of lesions, which is time-consuming and subjective. This study aimed to assess the feasibility of predicting muscle invasion in bladder cancer (BCa) with radiomics using a semi-automatic lesion segmentation method on T2-weighted images. Cases of non-muscle-invasive BCa (NMIBC) and muscle-invasive BCa (MIBC) were pathologically identified in a training cohort and in internal and external validation cohorts. For bladder tumor segmentation, a deep learning-based semi-automatic model was constructed, while manual segmentation was performed by a radiologist. Semi-automatic and manual segmentation results were respectively used in radiomics analyses to distinguish NMIBC from MIBC. An equivalence test was used to compare the models’ performance. The mean Dice similarity coefficients of the semi-automatic segmentation method were 0.836 and 0.801 in the internal and external validation cohorts, respectively. The area under the receiver operating characteristic curve (AUC) were 1.00 (0.991) and 0.892 (0.894) for the semi-automated model (manual) on the internal and external validation cohort, respectively (both *p* < 0.05). The average total processing time for semi-automatic segmentation was significantly shorter than that for manual segmentation (35 s vs. 92 s, *p* < 0.001). The BCa radiomics model based on semi-automatic segmentation method had a similar diagnostic performance as that of manual segmentation, while being less time-consuming and requiring fewer manual interventions.

## 1. Introduction

As the tenth most common type of cancer globally, bladder cancer (BCa) accounts for an estimated 500,000 new cases and 200,000 deaths annually [1,2]. BCa is divided into non-muscle-invasive BCa (NMIBC) and muscle-invasive BCa (MIBC) based on the depth and extent of invasion into the bladder wall as evaluated via biopsy, transurethral resection, and/or cystectomy [3,4]. NMIBC (urothelial carcinoma in situ, non-invasive papillary urothelial carcinoma and invasion into lamina propria) is typically treated by intravesical chemotherapy and/or bacillus Calmette–Guérin (BCG) immunotherapy or radical cystectomy (RC) [3,5]. By contrast, the recommended therapy for MIBC (invasion into muscularis propria and deeper) is radical cystectomy with lymph node dissection, selectively supplemented with chemotherapy, immunotherapy, or pembrolizumab [6]. The therapeutic selection relies on an accurate preoperative diagnosis of NMIBC or MIBC.

A repeated transurethral resection (TUR) is recommended if the first TUR for histological evaluation is incomplete, with upstaging to muscle-invasive disease in up to 45% of patients at stage T1 [7]. BCa cells, however, can be released into the bloodstream during TUR, increasing the risk of metastatic disease [8]. With high soft tissue contrast resolution, magnetic resonance imaging (MRI), a non-invasive and reproducible imaging modality, is capable of preoperatively staging and grading BCa [9,10]. Nevertheless, the visual inspection of MRI scans by radiologists is time-consuming, labor-intensive, subjective, and experience-dependent [11,12,13].

Radiomics has recently been applied in the diagnosis and prognosis of BCa and other cancers [14,15,16,17,18]. Radiomics in multiparametric breast MRI may have potential to predict human epidermal growth factor receptor 2 expression levels in breast cancer patients with distinct therapeutic ways [17]. In addition, a radiomics-based model can serve as a preoperative tool to predict macrophage infiltration and offer clinical guidance for immunotherapy of glioma [18]. Radiomics analysis involves the extraction of large amounts of feature data from radiographic images to establish a model to quantitatively describe tumor morphology and heterogeneity [11,19,20]. Feature extraction is accomplished by delineating the tumor on the images. Conventional radiomics analysis requires the manual delineation of lesions, and its accuracy is reliant on the experience of radiologists, as well as being labor-intensive [11]. There is a need for time- and labor-saving BCa lesion segmentation methods that also provide improved accuracy for differentiating between MIBC and NMIBC.

Computer-assisted segmentation is relatively reproducible and considerably less time- and labor-intensive. Deep learning-based automatic segmentation could be applied for epithelial ovarian cancer staging on T2-weighted MRI [21]. Ren et al. [22] proposed a novel approach for automatic segmentation of the prostate and its lesion regions on MRI with excellent performance. Their model was inspired by the U-Net model. Apart from cancer, the U-Net model could be used in the semantic segmentation of the urinary system from kidney, ureters, and bladder (KUB) X-ray images [23]. However, the accurate automatic segmentation of medical images remains a challenge, which can be attributed to the ambiguous, low-contrast, and heterogeneous boundary of the segmented area [24]. Several BCa radiomics segmentation methods have been shown to perform well [25,26,27,28]. The shape prior constrained particle swarm optimization (SPCPSO) model was proposed for automatic segmentation of the inner and outer boundaries of the bladder wall with good performance [25]. Li et al. [27] came up with a new model for bladder cancer segmentation based on high-throughput pixel-level features and a random forest (RF) classifier. This method only required a small database size to achieve a preferable segmentation performance compared with the conventional U-Net model. Our study utilized a two-dimensional U-Net network and a combination of the focal loss function and the soft Dice loss function, instead of a three-dimensional structure and categorical cross-entropy function in modelling, as in the work conducted by Coroama, D. M. et al. [28].

To this end, we aimed to assess the feasibility of predicting muscle invasion in BCa using radiomics analysis based on semi-automatic segmentation applied to T2-weighted (T2-W) MRI, compared to manual segmentation. Our hypothesis was that the diagnostic performance of the semi-automatic radiomics model would be equivalent to that of the manual method with an upper equivalence limit difference of AUC = 0.05.

## 2. Materials and Methods

### 2.1. Patients and Study Design

This retrospective multicenter study was approved by the Ethics Committee of of The Tenth Affiliated Hospital of Southern Medical University (Dongguan People’s Hospital) KYKT2021-044, and the requirement for patient consent was waived. From November 2019 to June 2022, 119 BCa patients with 160 tumors from the Tenth Affiliated Hospital of Southern Medical University (Dongguan People’s Hospital) (Center 1) and 54 BCa patients with 55 tumors from Sun Yat-Sen University Cancer Center (Center 2) were included in this study. The training and internal validation cohorts were obtained from Center 1, where images were acquired on a 3.0 Tesla Siemens MR scanner (MAGNETOM Skyra, Siemens, Erlangen, Germany), while the external validation cohort was obtained from Center 2, where images were acquired on a 3.0 Tesla United Imaging Healthcare MRI scanner (uMR 780, United Imaging Healthcare, Shanghai, China). All patients underwent bladder axial T2-weighted imaging in the supine position with a bladder filling of 250–300 mL. The MRI acquisition parameters used at the two centers are shown in Appendix A.

The inclusion criteria were as follows: (a) patients who were untreated or received only diagnostic transurethral resection of a bladder tumor; and (b) patients with BCa confirmed by radical or partial cystectomy or transurethral resection of bladder tumor within 2 weeks of undergoing multiparametric MRI. The exclusion criteria were as follows: (a) no surgical treatment and no pathological T staging; (b) histopathologically non-urothelial carcinoma; (c) tumor recurrence after BCa surgery; or (d) multiple tumors without an available histopathological diagnosis. The study design flow chart is displayed in Figure 1, including the flow chart of patient recruitment.

### 2.2. The Development and Validation of the Deep Learning-Based Semi-Automatic Segmentation Model

#### 2.2.1. Image Preprocessing

The tendency of bladder lesions to only occupy a small part of an MR image vastly increases the complexity of lesion segmentation. Each bladder tumor lesion was annotated by a radiologist (J.L., with 12 years of experience in uroradiology) with a cuboid box (volume of interest, VOI) using ITK-SNAP (version 3.4.0; www.itksnap.org, accessed on 1 June 2021). Thorough scrutiny was conducted on all VOIs to ensure the target BCa tumor lesion was completely included.

The voxel values of every VOI were reformatted as [0, 1] using min-max normalization to lower the data variance. To unify the input shape for the segmentation network, a zero-padding strategy was utilized to resample the VOI to 256 × 256 voxels in the xy-plane. Following normalization and resampling, each transverse slice within each VOI was fed into the segmentation network. The segmentation network used the contour of the lesion manually delineated layer-by-layer on the T2-W image by an experienced radiologist (J.L.) as the reference standard. The workflow of image preprocessing is shown in Figure 2A.

#### 2.2.2. Network Architecture

The segmentation network was a conventional deep learning-based architecture with a U-shaped encoder–decoder structure named U-Net [29], as depicted in Figure 2B. This architecture is commonly used for medical image segmentation due to its relatively simple structure, high efficiency, and ease of implementation. In the present study, the encoder phase was composed of five double-convolution (double-conv) blocks and four down-sampling blocks, while the decoder phase consisted of four double-conv blocks as well as four up-sampling blocks. Each double-conv block contained two convolution layers with a kernel of 3 × 3, and each layer was sequentially accompanied by a batch normalization layer and a rectified linear unit layer. A down-sampling block comprised a max-pooling layer with a kernel of 2 × 2 while an up-sampling block comprised a transpose convolution layer with a kernel of 4 × 4 and a concatenate layer which was used to merge the feature maps at the same depth. The input image size was reduced to 16×16 after encoding and reconstructed to 256 × 256 after decoding. The final layer of the network used a convolution layer with a kernel of 1 × 1 and a sigmoid function to produce a probability map with values between 0 and 1, comprising the segmentation output.

#### 2.2.3. The Development and Validation of the Semi-Automatic Segmentation Model

Model development involved combining the focal loss function [30] with soft Dice loss [31] to address the voxel class imbalance in segmentation. The model was trained on the training cohort using a cosine annealing strategy for decaying the learning rate from 1 × 10^−3^ to 1 × 10^−5^ and optimized by Adam [32] with a batch size of 20, a weight decay of 1 × 10^−4^, and an epoch of 100. The semi-automatic segmentation model was trained in the Python environment (version 3.6; https://www.python.org/, accessed on 1 November 2022) utilizing PyTorch (version 1.5.0; https://pytorch.org/, accessed on 1 November 2022) on a GPU NVIDIA GeForce GTX 1080TI installed on an Intel Xeon E5-2650 2.30 GHz×12 machine with the Linux Ubuntu 14.04 operating system.

The performance of the semi-automatic segmentation model was independently evaluated using the internal and external validation cohorts by comparing the segmentation results against the reference standard and calculating the Dice similarity coefficient (DSC), recall, and precision.

### 2.3. The Development and Validation of the Radiomics Model Based on Semi-Automatic Segmentation Results

#### 2.3.1. Feature Extraction and Selection

Feature extraction was performed in the Python environment (version 3.6; https://www.python.org/, accessed on 1 December 2022) using the pyradiomics toolkit (version 3.0.1; https://pyradiomics.readthedocs.io/en/latest/index.html, accessed on 1 December 2022). A total of 1130 radiomics features were extracted from the semi-automatic segmentation results of each bladder tumor lesion in the following categories: shape features, first-order features, texture features, and the features above based on different filters. The details of the extracted radiomics features are shown in Appendix A.

To eliminate redundant features, we first applied Z-score standardization to unify the units of each feature, followed by the least absolute shrinkage and selection operator (LASSO) algorithm to select features with non-zero coefficients. The feature selection procedure was executed on the training cohort and used for the validation cohorts, implemented in R (version 4.0.4; http://www.r-project.org/, accessed on 1 December 2022).

#### 2.3.2. The Development and Validation of the Radiomics Model

A binary classification model was constructed based on the selected features to differentiate between NMIBC and MIBC. The model utilized a support vector machine (SVM) [33] classifier with a radial basis function kernel [34]. To optimize the hyper-parameters, a grid search method was employed on the training cohort using the scikit-learn toolkit (version 19.0; https://scikit-learn.org/stable, accessed on 1 January 2023).

The diagnostic performance of the model was independently evaluated on the validation cohorts using receiver operating characteristic (ROC) analysis. The area under the ROC curve (AUC) and the corresponding 95% confidence interval (CI) were calculated. The accuracy, sensitivity, and specificity were calculated using the optimal cut-off value derived from maximizing the Youden index [35] from the ROC curve analysis.

### 2.4. The Development and Validation of the Radiomics Model Based on Manual Segmentation Results

To provide a comparator against which to assess the performance of using semi-automatic segmentation results to form a radiomics model for distinguishing NMIBC from MIBC, we utilized the method described in Section 2.3 to build a classification model based on manual segmentation results and train and validate it on the same data cohorts for comparison. It should be noted that the manual segmentation results were the lesion volumes delineated along the lesion contour by an experienced radiologist (J.L.), which were also the reference standard for semi-automatic segmentation.

To compare the time efficiency of the two methods, we counted the time taken to generate semi-automatic segmentation results (including image preprocessing and segmentation model prediction) and the amount of time taken by the radiologist to manually perform the segmentation.

### 2.5. Statistical Analysis

Statistical analyses were implemented in R (version 4.0.4; http://www.r-project.org/, accessed on 1 March 2023) and MedCalc (version 15.8; http://www.medcalc.org/, accessed on 1 March 2023).

The sample size required to detect an AUC value different from 0.500 was estimated with the following parameters: power, 80%; two-sided significance level, 0.05; alternative hypothesis of the true AUC values of the radiomics model based on semi-automatic segmentation results in internal and external validation cohorts compared with the null hypothesis of AUC = 0.5; the ratio of classes, the real ratios in our study (internal validation cohort = 27NMIBC/4MIBC, external validation cohort = 23NMIBC/32MIBC) [36,37].

The Kolmogorov–Smirnov test or Shapiro–Wilk *W* test were used to test whether the measurement data were normally distributed. Student’s *t*-test was used for continuous variables with normal distribution, expressed as a mean ± standard deviation (SD), while the Mann–Whitney *U* test was used for non-normally distributed variables, presented as a median (upper quantile to lower quantile). The chi-square test was applied for categorical variables, expressed as a ratio.

DeLong’s test was utilized to compare the AUCs of the different radiomics models. An equivalence test, calculated by confidence interval, of the difference in AUCs [38] was applied to verify whether the diagnostic performance of the radiomics models was equivalent, with a limit of AUC = 0.05, as recommended by Zhou et al. [39]. To explore the reason for any difference in the diagnostic performance of the two radiomics models, we calculated the intraclass correlation coefficients (ICCs) of the radiomics features extracted from the respective segmentation results across the entire data cohort.

A two-tailed *p* value less than 0.05 was considered statistically significant.

## 3. Results

### 3.1. Patient Characteristics

This study included 94 patients with 129 tumor lesions (103 NMIBC tumors, 26 MIBC tumors) in the training cohort, 25 patients with 31 tumor lesions (27 NMIBC tumors, 4 MIBC tumors) in the internal validation cohort, and 54 patients with 55 tumor lesions (23 NMIBC tumors, 32 MIBC tumors) in the external validation cohort. The demographic characteristics of all cohorts are shown in Table 1, with no significant differences in age and sex between cohorts.

### 3.2. Performance Evaluation of the Deep Learning-Based Semi-Automatic Segmentation Model

The results of semi-automatic segmentation for the internal and external validation cohorts are shown in Table 2. The mean DSC in the internal validation cohort was 0.836 ± 0.085, while the mean recall and precision were 0.803 ± 0.115 and 0.885 ± 0.072, respectively. The mean DSC in the external validation cohort was 0.801 ± 0.112, and the mean recall and precision were 0.782 ± 0.119 and 0.838 ± 0.140, respectively. Images and segmentations of four typical bladder lesions are provided in Figure 3.

### 3.3. Performance Evaluation of the Radiomics Model Based on Semi-Automatic Segmentation Results

#### 3.3.1. Radiomics Feature Selection

Of the 1130 radiomics features extracted from each VOI, 12 features with non-zero coefficients were ultimately retained for radiomics modeling according to LASSO, with an optimal lambda (λ) value of 0.034 (ln⁡λ=−3.381; Figure 4).

The details of the 12 selected features and their coefficients are shown in Table 3. For the training and validation cohorts, all but three of the selected features significantly differed between NMIBC and MIBC. For the remaining nine features, *kurtosis*, the feature with the largest coefficient, and *skewness*, the feature with multiple occurrences, were first-order features. The values of the *skewness* feature for NMIBC were significantly lower than those for MIBC across all cohorts (both *p* < 0.001; Table 3). Figure 5A,B show images of representative patients with NMIBC and MIBC, respectively, on T2-W images and relevant feature maps overlaid on the transverse section. As shown in the *skewness* feature maps (Figure 5(Af,Ag,Bf,Bg)), NMIBC shows relatively uniform speckles, while MIBC tends to show localized asymmetries. The value of the *sphericity* feature for MIBC was smaller than that for NMIBC (Table 3).

#### 3.3.2. Classification Performance in the Validation Cohorts

The classification performance of the radiomics model based on semi-automatic segmentation results in the internal and external validation cohorts is shown in Table 4. The radiomics model reached an AUC of 1.000 (95% CI, 0.888–1.000) in the internal validation cohort and an AUC of 0.892 (95% CI, 0.779–0.960) in the external validation cohort (both *p* < 0.001; Figure 6). 

Sample size calculations were used to determine that a sample of twelve tumor lesions (two with MIBC and ten with NMIBC) was required for internal validation, and a sample of fifteen tumor lesions (nine with MIBC and six with NMIBC) was required for external validation. Hence, the sample sizes in this study (31 and 55 in the internal and external validation cohorts, respectively) were sufficient for assessing that the true AUCs of 1.000 and 0.892 were different from an AUC of 0.500 with 80% power.

### 3.4. Comparison of the Radiomics Models Based on Different Segmentation Results

The feature selection results of the radiomics model based on manual segmentation are shown in Appendix A, wherein 24 radiomics features are ultimately selected. The model achieved a similar performance to that of the model based on semi-automatic segmentation results, with an AUC of 0.991 (95% CI, 0.871–1.000; *p* < 0.001) in the internal validation cohort and 0.894 (95% CI, 0.781–0.961; *p* < 0.001) in the external validation cohort (Table 4; Figure 6).

DeLong’s test showed there was no significant difference in AUCs between the two radiomics models either in the internal validation cohort (AUC = 1.000 vs. 0.991, *p* = 0.480) or the external validation cohort (AUC = 0.892 vs. 0.894, *p* = 0.930). Meanwhile, the differences between the AUCs of the two models were 0.009 (95% CI, −0.016 to 0.035) and 0.002 (95% CI, −0.044 to 0.048) in the internal and external validation cohorts, respectively (Table 4). Both 95% CIs of differences in AUCs fell entirely within the zone of equivalence (interval ± 0.05) in the internal and external validation cohorts. Therefore, the diagnostic performance of the radiomics model based on semi-automatic segmentation for distinguishing NMIBC from MIBC was equivalent to that of the model based on manual segmentation, as demonstrated by the equivalence test with a limit of AUC = 0.05. Moreover, seven cases were misclassified in both models under the setting thresholds—two as false positives and five as false negatives, and no significant difference in accuracy was observed between the two methods (0.818 vs. 0.800, *p* > 0.999), according to McNemar’s test.

Of the 1130 radiomics features extracted based on the two different segmentation results, respectively, the median ICC was 0.988 (interquartile range, 0.968 to 0.995), suggesting high consistency. Notably, of the features that were ultimately selected for the two models, eight features overlapped, with all ICCs > 0.9.

The processing time for VOI annotation in the semi-automatic method was 34 s (17–50 s) and the total time for generating a semi-automatic segmentation result (including image preprocessing and segmentation model prediction) was 35 s (18–52 s). The radiologist took 92 s (30–356 s) to manually delineate the images, which indicates a statistically significant reduction in processing time (35 s vs. 92 s, *p* < 0.001).

## 4. Discussion

We proposed a semi-automatic segmentation method based on T2-W MRI for radiomics analysis to predict muscle invasion in BCa. With shorter processing times, the radiomics model based on semi-automatic segmentation results displayed a similar prediction performance to that based on manual segmentation results.

The radiomics model based on semi-automatic segmentation exhibited good segmentation performance in both the internal and external validation cohorts, with DSCs of 0.836 and 0.801. Similarly high DSCs have also been demonstrated in other studies. Dong et al. [40] used U-Net for BCa segmentation and obtained a DSC of 0.84. Moribata et al. [41] developed a versatile automatic segmentation model for BCa using apparent diffusion coefficient (ADC) maps calculated from diffusion MRI and achieved an average DSC of 0.83. Thus, the use of U-Net has been demonstrated to be able to capture sufficient information about tumor boundaries and extract effective features through its U-shaped encoder–decoder structure and unique skip-connection mechanism, resulting in a relatively high segmentation accuracy [11]. The crucial discriminator between MIBC and NMIBC is whether the peripheral part of the lesion invades the muscularis propria of the bladder wall. The microscopic invasiveness of BCa cells in the boundary poses a challenge for the accurate segmentation of BCa lesions. Models using U-Net structures may address this problem to some extent.

Our semi-automatic segmentation results were stable, with high precision in the internal and external validation datasets, in agreement with a previous study using multi-input dilated U-shape neural networks for the segmentation of BCa [42]. The stability of semi-automatic segmentation results in the present study may be explained by the combined use of focal loss and soft Dice loss functions in the development of the model, which led the model to focus on the tumor lesions despite the voxel-wise distribution of tumor lesions and the surrounding non-tumor tissues being unbalanced.

Feature selection played a critical role in enhancing the classification performance of the radiomics model. The selected features of the semi-automatic segmentation-based radiomics model were favorable for discriminating between MIBC and NMIBC. Among the radiomics features that statistically differed between MIBC and NMIBC, *kurtosis*, the feature with the largest coefficient, and *skewness*, the feature with multiple occurrences, were first-order features which described the gray-level distribution within the VOI. The *kurtosis* feature reflected the sharpness of the histogram distribution and the tissues’ microstructure, while the *skewness* feature represented the asymmetry of histogram distribution [43]. The values of the *skewness* feature for MIBC were significantly higher than those for NMIBC across all data cohorts, demonstrating a higher heterogeneity of MIBC in comparison to NMIBC. The *skewness* feature has also been used to distinguish the molecular characterization of lung adenocarcinoma [44] and invasive ductal breast cancer [45] in previous radiomics analyses. Further, the *sphericity* feature (a shape feature), which had the second largest coefficient, described the sphericity of a given region and reflected the closeness of a three-dimensional shape of the VOI to the sphere. The value of the *sphericity* feature for MIBC was smaller than that for NMIBC, indicating that the shape of the VOI in MIBC was much less regular, while reflecting the heterogeneity of MIBC. The *skewness* and *sphericity* features used in the proposed radiomics model may serve as biomarkers for differentiation between MIBC and NMIBC.

The diagnostic capacity of the radiomics model based on semi-automatic segmentation for predicting muscle invasion was not significantly different from that of the model based on manual segmentation. ICCs showed that the radiomics features selected by the two models were highly consistent (with eight overlapping features, ICCs > 0.9), leading to a similar classification performance. Our study showed that diagnostic performance may not depend on a high DSC, even though the DSC of our semi-automatic segmentation method was comparatively lower than that reported in previous studies [26,27,46]. The semi-automatic analysis method also achieved good performance when predicting the pathological grading of pancreatic neuroendocrine neoplasms (pNENs) [11], with no statistical difference between AUCs for manual and semi-automatic segmentation, consistent with the present study. In addition, a deep learning-based classification method has been used in a previous study to directly predict muscle invasion in bladder cancer [13], which did not require segmentation but had a slightly lower AUC than that achieved in this study in the same internal and external validation cohorts (AUC = 0.963 and 0.861, respectively). Therefore, the proposed radiomics model based on semi-automatic segmentation results could be considered to have the most promising predictive capability.

Considering that the experience levels of radiologists will vary, leading to differing levels in the accuracy of and time taken to complete BCa lesion VOI contouring, we chose one experienced radiologist to complete the lesion annotations of all patients included in this study. However, the processing time for VOI annotation of different tumor lesions still varied dramatically, which may partly be attributed to the heterogeneity of BCa and partly due to the radiologist’s unfamiliarity with the image format used in the external validation cohort. However, importantly, the total time required for semi-automatic segmentation was significantly less than that required for manual segmentation. While saving significant labor and time, the radiomics model based on semi-automatic segmentation achieved a similar differentiation performance between MIBC and NMIBC to that based on manual segmentation. The robustness of our semi-automatic segmentation method indicated that it has the potential to serve as a prognostic tool for the prediction of BCa pathological invasion preoperatively.

The limitations of our study were as follows. Firstly, the radiomics analysis based on semi-automatic image segmentation still required some manual input to identify the location of the lesion(s). Achieving a completely automatic segmentation method will require further development but will significantly improve the efficiency of the total analysis. Secondly, tumor segmentation was accomplished using T2-W imaging alone. Including additional image contrast mechanisms or multiparametric MRI may improve the performance of the model but may also increase its complexity. Finally, the relatively small sample size and inclusion of only two centers may have introduced bias and decreased the generalizability of the semi-automatic segmentation model.

## 5. Conclusions

We developed and validated a radiomics model based on semi-automatic segmentation of T2-W MRI for the prediction of muscle invasion in BCa. With increased speed and automation, the radiomics model based on the semi-automatic segmentation method showed a similar diagnostic performance to that of the manual segmentation method. The method we proposed may have the potential to serve as a prognostic tool for the preoperative prediction of BCa pathological invasion.

## Figures and Tables

**Figure 1 bioengineering-10-01355-f001:**
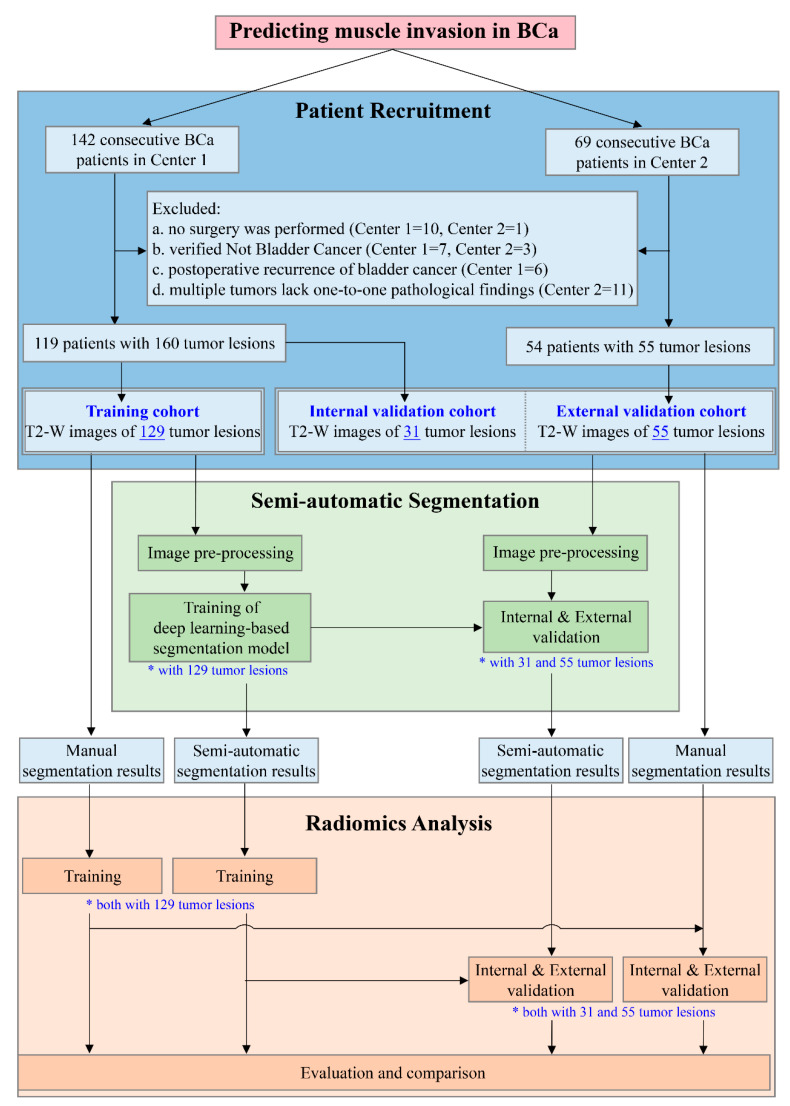
Study design flowchart. A total of 119 patients with 160 bladder tumor lesions in Center 1 and 54 patients with 55 tumor lesions in Center 2 were included. The training cohort, consisting of 129 tumor lesions from Center 1, was used to develop the semi-automatic segmentation model and the segmentation results were subsequently employed for the development of the radiomics model. The remaining 31 tumor lesions from Center 1 and 55 tumor lesions from Center 2 were regarded as the internal and external validation cohorts, respectively, for the independent evaluation of each model to compare the performance of the radiomics model based on semi-automatic segmentation with that of the radiomics model based on manual segmentation. Both * and underline highlight the number of tumor lesions in the corresponding procedure. (BCa, bladder cancer; T2-W, T2-weighted).

**Figure 2 bioengineering-10-01355-f002:**
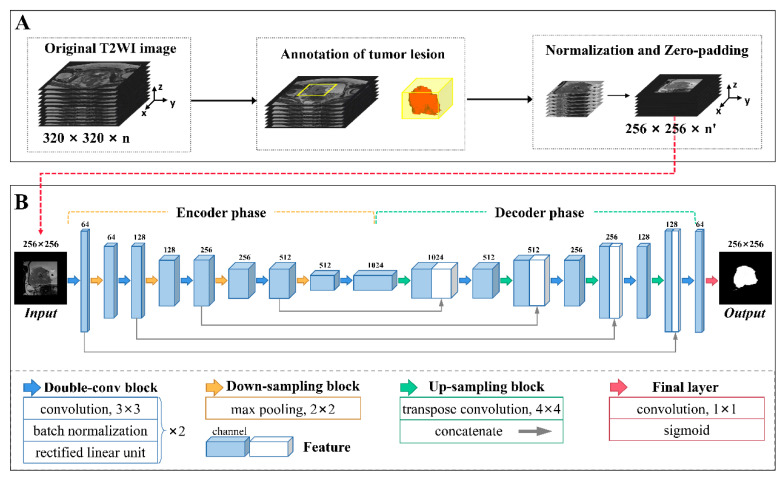
The development of a deep learning-based semi-automatic segmentation model. The procedure broadly comprised two parts: (**A**) image preprocessing and (**B**) network architecture. (**A**) The workflow of image preprocessing consisted of tumor lesion annotation, normalization, and resampling to 256 × 256 with zero padding. (**B**) The architecture of the U-Net network. Differently colored arrows indicate network operations, as shown at the bottom. Cuboids represent the feature maps after operations. The number above each cuboid represents the channel number of the feature map.

**Figure 3 bioengineering-10-01355-f003:**
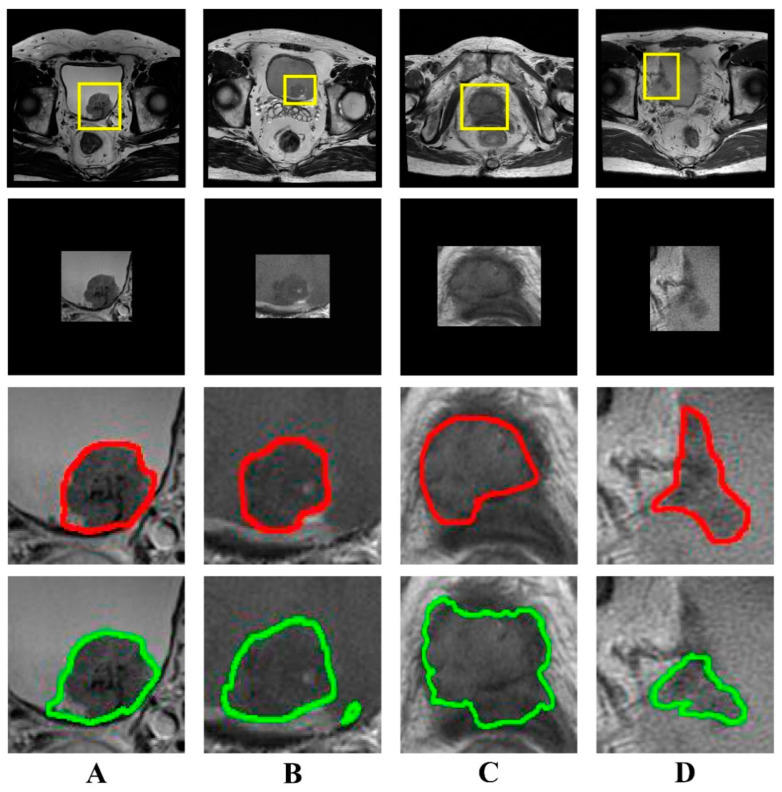
Examples of bladder lesion segmentation. The DSC of each case (**A**–**D**) was 0.945, 0.861, 0.752, and 0.668, respectively. The first row shows T2-weighted images (320 × 320) with radiologist-identified tumor regions (yellow rectangle); the second row shows the images (256 × 256) of tumor regions with zero padding for input into the U-Net network; the third row shows the reference standard drawn by the radiologist (red line); the fourth row shows the output of the semi-automatic segmentation model (green line). (DSC, Dice similarity coefficient).

**Figure 4 bioengineering-10-01355-f004:**
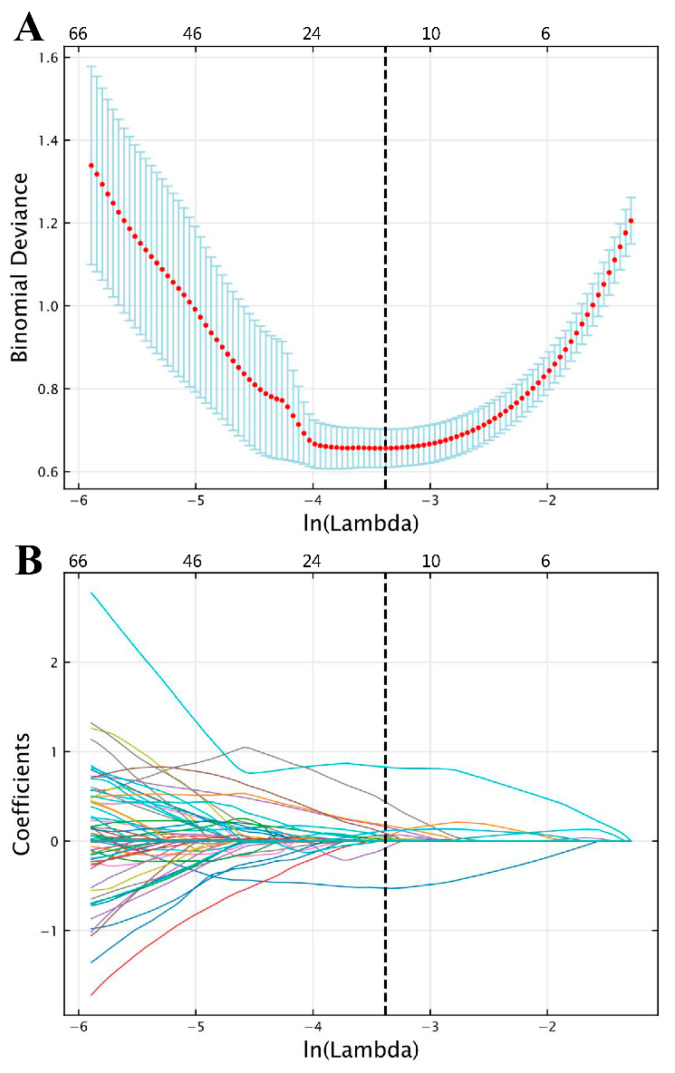
Feature selection results following LASSO based on features extracted from semi-automatic segmentation results. (**A**) Selection of LASSO model’s tuning parameter lambda (λ), based on 10-fold cross-validation using minimum criteria. The vertical dashed line indicates the optimal value of λ. (**B**) Coefficient profile plot. Each colored line represents the coefficient of each radiomics feature, and the vertical dashed line represents 12 radiomics features with non-zero coefficients selected with optimal λ. In each plot, the x-axis at the bottom shows ln⁡(λ), while the x-axis at the top shows the number of remaining radiomics features that vary with λ. (LASSO, least absolute shrinkage and selection operator).

**Figure 5 bioengineering-10-01355-f005:**
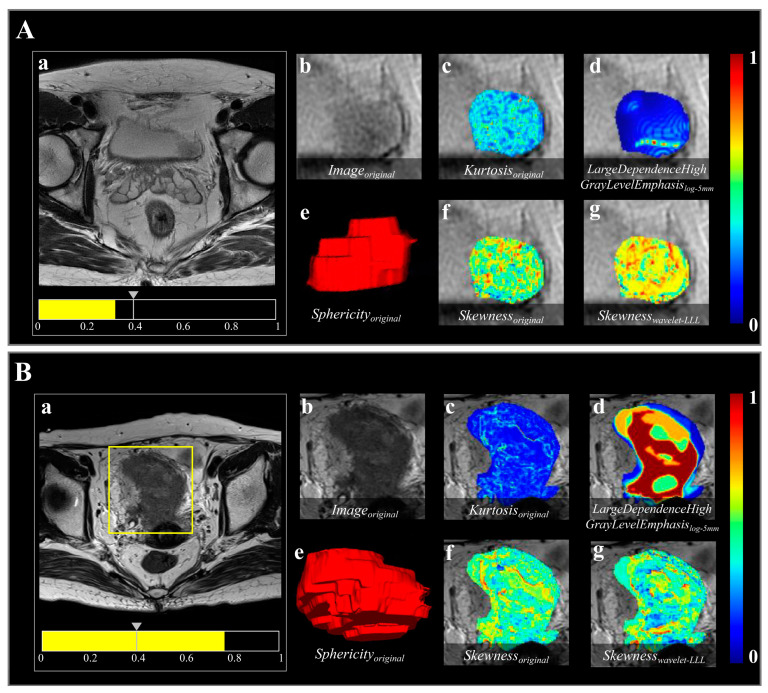
Two representative bladder lesions analyzed using the radiomics model based on semi-automatic segmentation results. Images show the result in (**A**) a 63-year-old male with NMIBC and (**B**) a 76-year-old female with MIBC. The predicted probabilities of the radiomics model for the cases in (**A**,**B**) were 0.321 and 0.769, respectively. (**a**) A threshold of 0.396 (the gray arrow on the scale) was used to visualize the lesion in each case and for the model to classify them as either NMIBC or MIBC. (**b**) shows the region highlighted in the yellow box in (**a**), indicating the tumor lesion in the original T2-W image. (**c**–**g**) Radiomics features overlaid on the T2-W image, as follows: (**c**) *Kurtosis_original_*, first-order feature indicating sharpness of gray-level histogram distribution; (**d**) *LargeDependenceHighGrayLevelEmphasis_log-5mm_*, texture feature representing gray-level dependence matrix (gldm) achieved through transformation of “Laplacian of Gaussian (LoG)” filter with sigma value of 5.0, indicating intratumoral heterogeneity; (**e**) *Sphericity_original_*, shape feature, volume-rendering reconstruction of bladder lesion indicating three-dimensional shape; and (**f**,**g**), *Skewness_original_* and *Skewness_wavelet-LLL_*, both first-order features, indicating asymmetry of gray-level histogram distribution, obtained from image transformed by “Wavelet” filter with three high-pass [H] filters acting on x, y, and z axes (LLL).

**Figure 6 bioengineering-10-01355-f006:**
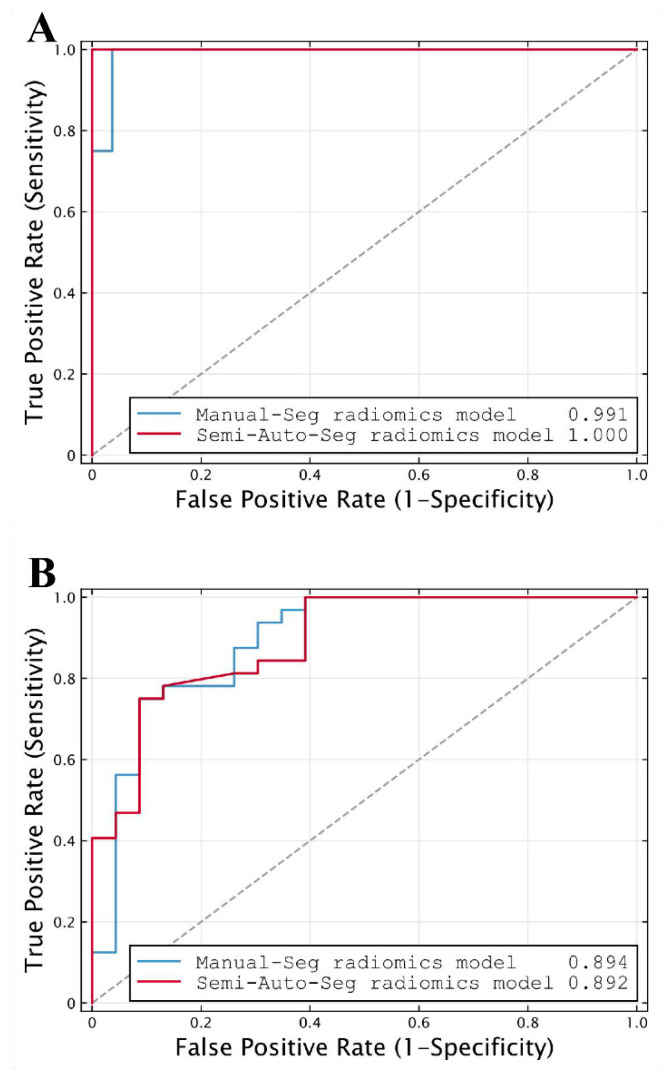
ROC curves of radiomics models based on manual segmentation results (Manual-Seg radiomics model) and radiomics models based on semi-automatic segmentation results (Semi-Auto-Seg radiomics model) in (**A**) internal and (**B**) external validation cohorts, used for distinguishing NMIBC from MIBC. The dot line represents an AUC of 0.5. (ROC, receiver operating characteristic; AUC, the area under the receiver operating characteristic curve).

**Table 1 bioengineering-10-01355-t001:** Patient characteristics.

Characteristics	Training Cohort	Internal Validation Cohort		External Validation Cohort	
	Center 1	Center 1	*p*	Center 2	*p*
Number, n	129	31	-	55	-
Age (years)			0.553 ^†^		0.244 ^†^
Median (IQR)	67 (59, 75)	69 (62, 71)		64 (55, 73)	
Sex			0.561 ^‡^		0.869 ^‡^
Male	109	31		47	
Female	20	0		8	
MRI-determined tumor size (cm)			0.892 ^†^		<0.001 ^†^
<3	100	25		12	
≥3	29	6		43	
Pathological T stage			-		-
Ta	75	23		20	
T1	28	4		3	
T2	15	2		15	
T3	4	2		13	
T4	7	0		4	
Pathological grade			-		-
Low	51	16		10	
High	78	15		45	
Degree of infiltration			-		-
NMIBC	103	27		23	
MIBC	26	4		32	

Note. NMIBC, non-muscle-invasive bladder cancer; MIBC, muscle-invasive bladder cancer. ^†^ *p* value is the significance level of comparison of the difference between the training cohort and the internal/external validation cohorts according to the Mann–Whitney U test. ^‡^ *p* value is the significance level of comparison of the difference between the training cohort and the internal/external validation cohorts by chi-square test.

**Table 2 bioengineering-10-01355-t002:** Performance of the deep learning-based semi-automatic segmentation model in internal and external validation cohorts.

Dataset	DSC			Recall		Precision	
	Mean ± SD	Median	Range	Mean ± SD	Median	Mean ± SD	Median
Internal Validation Cohort (n = 31 ^†^)	0.836 ± 0.085	0.861	0.558–0.921	0.803 ± 0.115	0.845	0.885 ± 0.072	0.900
External Validation Cohort (n = 55 ^†^)	0.801 ± 0.112	0.841	0.525–0.951	0.782 ± 0.119	0.821	0.838 ± 0.140	0.880

Note. DSC, Dice similarity coefficient; SD, standard deviation. DSC, recall and precision of the semi-automatic segmentation model in internal and external validation cohorts were calculated with a cut-off value of 0.5. ^†^ Number of tumor lesions involved.

**Table 3 bioengineering-10-01355-t003:** Selected radiomics features of the model based on semi-automatic segmentation for classifying NMIBC and MIBC.

Feature Name	Coefficient	Training Cohort			Internal Validation Cohort			External Validation Cohort		
		NMIBC	MIBC	*p **	NMIBC	MIBC	*p **	NMIBC	MIBC	*p **
original_firstorder_Kurtosis	0.129	−0.398 (−0.637, 0.060)	0.763 (−0.085, 1.974)	<0.001	−0.411 (−0.628, −0.245)	1.787 (1.125, 3.313)	<0.001	−0.027 (−0.422, 0.235)	0.730 (0.037, 1.370)	<0.001
original_shape_Sphericity	−0.077	0.307 (−0.219, 0.805)	−0.471 (−1.195, −0.085)	<0.001	0.439 (−0.492, 0.704)	−1.366 (−1.619, −1.109)	0.005	−0.340 (−1.106, 0.168)	−2.318 (−3.933, −1.116)	<0.001
original_firstorder_Skewness	0.048	−0.232 (−0.842, 0.201)	1.316 (0.389, 1.834)	<0.001	0.037 (−0.670, 0.363)	1.962 (1.403, 2.433)	<0.001	−0.179 (−0.698, 0.317)	0.877 (0.149, 1.196)	<0.001
log-sigma-5-0-mm-3D_gldm_LargeDependenceHighGrayLevelEmphasis	0.032	−0.512 (−0.697, −0.011)	0.667 (−0.086, 2.005)	<0.001	−0.273 (−0.450, −0.109)	1.781 (1.354, 3.022)	<0.001	0.298 (0.019, 1.243)	1.933 (1.049, 4.171)	<0.001
log-sigma-1-0-mm-3D_ firstorder_Skewness	0.028	−0.105 (−0.590, 0.630)	−0.123 (−0.669, 0.426)	0.787	0.071 (−0.700, 0.377)	−0.112 (−0.912, 0.072)	0.589	−0.439 (−1.075, −0.100)	0.351 (−0.120, 1.177)	<0.001
wavelet-LLL_firstorder_ Skewness	0.020	−0.310 (0.751, 0.265)	1.247 (0.442, 1.837)	<0.001	0.002 (−0.538, 0.243)	1.860 (1.210, 2.433)	<0.001	−0.199 (−0.588, 0.070)	0.787 (0.107, 1.205)	<0.001
log-sigma-1-0-mm-3D_glcm_ClusterShade	0.014	0.110 (−0.076, 0.159)	0.151 (0.066, 0.188)	0.040	0.121 (−0.304, 0.180)	0.146 (0.034, 0.245)	0.589	−0.136 (−0.633, 0.091)	0.247 (0.139, 0.584)	<0.001
wavelet-HHL_glcm_MCC	−0.009	−0.222 (−0.687, 0.531)	−0.514 (−0.826, 0.292)	0.080	−0.223 (−0.496, 0.636)	−0.135 (−0.679, 0.702)	0.932	−0.260 (−0.476, 0.431)	−0.446 (−0.623, 0.011)	0.091
original_shape_ MinorAxisLength	0.007	−0.453 (−0.750, 0.027)	0.451 (0.044, 1.915)	<0.001	−0.285 (−0.593, −0.056)	1.591 (0.778, 2.108)	<0.001	0.075 (−0.524, 0.408)	1.181 (0.483, 1.992)	<0.001
wavelet-HLH_glszm_ GrayLevelNonUniformity	0.004	−0.364 (−0.403, 0.205)	−0.097 (−0.230, 0.580)	<0.001	−0.310 (−0.373, −0.227)	1.165 (0.444, 2.328)	<0.001	0.111 (−0.281, 0.698)	1.636 (0.749, 4.001)	<0.001
log-sigma-5-0-mm-3D_gldm_DependenceNonUniformity	0.002	−0.394 (−0.452, −0.267)	−0.043 (−0.268, 1.126)	<0.001	−0.379 (−0.433, −0.245)	0.972 (0.260, 2.074)	<0.001	0.183 (−0.329, 0.717)	1.829 (0.448, 4.083)	<0.001
log-sigma-5-0-mm-3D_glszm_SmallAreaEmphasis	0.002	−0.248 (−1.035, 0.377)	0.659 (0.357, 1.002)	<0.001	−0.294 (−0.898, 0.863)	1.079 (0.832, 1.310)	0.039	0.094 (−0.238, 0.450)	0.630 (0.283, 0.984)	0.001

Note. Each feature is denoted by concatenating the image type, feature group, and feature name by underline. For example, original_firstorder_Kurtosis is a feature extracted from the original image, belongs to the first-order group, and the feature name is *kurtosis*. All radiomics feature values are those after Z-score standardization and presented as median (upper quantile to lower quantile). Glcm, gray-level co-occurrence matrix; glszm, gray-level size zone matrix; gldm, gray-level dependence matrix; all features above belong to texture features. * *p* values indicate comparisons of the radiomics feature value difference between NMIBC and NIBC in each data cohort according to Mann–Whitney U test.

**Table 4 bioengineering-10-01355-t004:** Diagnostic performance of radiomics models based on semi-automatic and manual segmentation results.

Dataset	Segmentation Method	Accuracy	Sensitivity	Specificity	AUC (95% CI)	AUC Difference (95% CI)	*p*	
Internal validation cohort (MIBC/NMIBC = 4/27)	Semi-automatic	1.000	1.000	1.000	1.000 (0.888–1.000)	0.009 (−0.016–0.035)	<0.001 ^†^	0.480 ^‡^
	Manual	0.968	1.000	0.963	0.991 (0.871–1.000)		<0.001 ^†^	
External validation cohort (MIBC/NMIBC = 32/23)	Semi-automatic	0.818	0.750	0.913	0.892 (0.779–0.960)	0.002 (−0.044–0.048)	<0.001 ^†^	0.930 ^‡^
	Manual	0.800	0.750	0.870	0.894 (0.781–0.961)		<0.001 ^†^	

Note. AUC, area under receiver operating characteristic curve; CI, confidence interval. Accuracy, sensitivity, and specificity of radiomics model based on semi-automatic and manual segmentation results in internal and external validation cohorts were calculated with cut-off values of 0.396 and 0.574, respectively. ^†^
*p* value is the significance level of comparison of AUC with that of random case (AUC = 0.5). ^‡^ *p* value is the significance level of comparison of AUCs between radiomics models based on different segmentation methods (semi-automatic vs. manual) according to DeLong’s test.

## Data Availability

Data are not publicly available, but authors can be contacted privately if requested.

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
