# Peer review of "Radiomics Prediction of Muscle Invasion in Bladder Cancer Using Semi-Automatic Lesion Segmentation of MRI Compared with Manual Segmentation"

_bioengineering, 2023, doi:10.3390/bioengineering10121355_

Round 1
Reviewer 1 Report
Comments and Suggestions for Authors
In this article, the authors outline a novel procedure to predict contours around lesions in order to identify the severity of the muscle invasion in bladder cancer patients. The model is semi-automated and consists of two parts, the first being a deep-learning based algorithm to properly segment the area of lesion, thus potentially speeding up a time-consuming and laborious process and then finally using that segmented region to identify whether the lesion is non muscle-invasive (NMIBC) or muscle-invasive (MIBC) determining different therapeutic treatments. The analysis is well designed and care is taken to identify before-hand potential biases in the training sample. The model is adequately tested on two different samples and conclusions are drawn carefully with sound statistical judgement. The value of the procedure is also obvious and I would very much recommend the article as suitable for publication in this journal.
Please find below some specific comments :
Abstract "The area under the receiver operating characteristic curve .. "
- Suggest adding the AUC numbers of the semi-automated model along with the manual method like "... (AUC=1.00 (0.991) and 0.892 (0.894)) for the semi-automated model (manual) on the internal and external validation cohort respectively". Otherwise it reads like the 0.991 AUC is from the manual model and 0.894 from the semi-automated model raising questions about the difference in performance.
- Please add missing references for SVM and RBF Kernel (for eg Cortes, C., & Vapnik, V. (1995). Support-vector networks) as well as the Youden Index in Section 2.3.2
- Table 2 and the AUC numbers in Table 4 for eg. suggest slightly weaker but noticeable drop-off in performance for the external validation cohort vs internal validation cohort. On it's face this might not be too surprising since the external sample has a lot more MIBC proportionally compared to the training and internal validation sample. Is there a way to check that this is indeed the case? The efforts to remove class composition bias using a focal loss function is encouraging. Was there a training done for eg without this to see if the performance on the external sample became worse?
- Fig 3 (bottom row, B column) raises the interesting question about the model sometimes predicting multiple disconnected regions of lesions. In general, I imagine this is not desired. How frequent was this occurrence? Is there a way to enforce this in the U-net architecture to ensure only a single contour is drawn?
- The numbers in table 4 showing how similar the two different methods are is encouraging. However, since the samples being looked at under these methods are the exact same, could the authors also provide an indication of whether the inaccuracies are mostly shared? For eg, how many of the 55 samples in the external cohort are classified inaccurately by both methods vs how many are uniquely by one method or the other. This would further test whether the methods are using similar features in the samples to make a decision.
Reviewer 2 Report
Comments and Suggestions for Authors
The authors are presenting a new semi-automated region segmentation of MRI images in the context of bladder cancer.
The paper is well written and deserve, in my opinion, to be published due to the interest for the scientific community.
I only have few minor comments.
Introduction :
Since the manuscript is also for non-oncologist public, would it be possible to remind the different stages of bladder cancer ?
I suggest to mention the use of radiomics for other types of cancer (and had references)
Figure 5 :
What do you mean by " (a kind of texture feature ) "?
I would prefer sharpness instead of “pointedness and peakedness”. (but you can keep it this way, it’s up to you)
Since you are giving a brief definition of kurtosis you should give one for skewness (it only comes at the end in the discussion)
In supplementary :
S1 : What type of sequence do you use ? Since it is T2-W images I suppose it is spin echo sequence, in this case it is not relevant to mention the flip angle.
Where does come from the "Number of excitations of 1.5" ? I'm use to discret values, does 1.5 comes from the consideration of an acceleration factor ?
Reviewer 3 Report
Comments and Suggestions for Authors
This paper developed a method using semi-automatic lesion segmentation on MRI images to predict muscle invasion in bladder cancer. The paper compared the results from the manual segmentation and the semi-automatic segmentation, and the semi-automatic method achieved better accuracy with a shorter time and less manual intervention. The following concerns need to be addressed:
- How is this work different from other works related to using the automatic method for bladder tumor segmentation and bladder cancer diagnosis (e.g.Fully automated bladder tumor segmentation from T2 MRI images using 3D U-Net algorithm, doi: 10.3389/fonc.2023.1096136. )?
- The author only mentioned briefly in the Introduction, "Several BCa radiomics segmentation methods have been shown to perform well [19-21]. " . The author should add more background on these works that used automatic segmentation for cancer diagnosis in the Introduction.
- The paper compared (1) manual segmentation + radiomics analysis and (2) U-Net segmentation + radiomics analysis. If the U-Net segmentation is highly accurate, meaning it completely matches manual label results, the classification results between the two methods should be very close. Considering this, it is more interesting to add a comparison group (3) no segmentation, which uses a deep learning-based method to predict muscle invasion in bladder cancer directly.
- What is the difference between Figure S1 and Figure 4. Why does the paper use 12 radiomics features instead of 24 features?
Reviewer 4 Report
Comments and Suggestions for Authors The presented paper provides a strong background to the methods used for semi-automated segmentation and radiomics analysis. In order to gain a more complete picture of the study and the issues that the authors have considered while conducting this research, some aspects must be delved into in greater detail. I report the aspects on which I invite the authors to focus: 1) Can more information be provided about the age, sex, stage of bladder cancer, among other characteristics? This might also explain the study population and sample representation. 2) Would it also be possible to integrate other imaging modalities, such as MRI into predictive model?3) What would be the result if additional variables were added in this model?
4) It would be useful in the related work phase to refer to recent contributions that propose techniques for the semantic segmentation of the urinary system (see KUB-UNet: Segmentation of urinary system organs from KUB x ray image).
5) What are major drawbacks of using a semi-automatic segmentation in the project? Are there any flaws or biases that could affect your conclusions?
6) What were your strategies for minimizing possible errors and biases in analyzing the data? What sensitivity analysis did you carry out or what other ways were used to determine if the results are robust? Nowadays, the scientific community turns to such solutions as argumentation frameworks which are meant to be more comprehensible (see Argumentation approaches for explainable AI in medical informatics).
7) Have you evaluated the efficiency in terms of cost and compared with alternative procedures? What are the possible implications of your approach on existing clinical practices as well as the financial aspects involved in the diagnosis and treatment of bladder cancer?
8) How important are the findings in your study from a practical point of view? What is the potential impact of your results on bladder cancer diagnostic and therapeutic practices?
9) Where might research be headed in the future? Can you point out any areas that could be explored further so as to enhance understanding and therapy of bladder cancer?
Comments on the Quality of English Language
Fine
Round 2
Reviewer 3 Report
Comments and Suggestions for Authors
The author has addressed my comments.
Reviewer 4 Report
Comments and Suggestions for Authors The authors followed the suggestions proposed. The paper can be published